# Intensive community and home-based treatments for eating disorders: a scoping review study protocol

Başak İnce  ,[1] Matthew Phillips,[1] Ulrike Schmidt[1,2]

**To cite:** İnce B, Phillips M, Schmidt U. Intensive community and home-based treatments for eating disorders: a scoping review study protocol. *BMJ Open* 2023;13:e064243. doi:10.1136/bmjopen-2022-064243

[1]Section of Eating Disorders, Department of Psychological Medicine, Institute of Psychiatry Psychology and Neuroscience, King's College London, London, UK
[2]Eating Disorder Outpatients Service, South London and Maudsley NHS Foundation Trust, London, UK

**Correspondence to**
Dr Başak İnce;
basak.ince@kcl.ac.uk

## ABSTRACT

**Introduction** Institutionally based intensive treatment modalities (inpatient, day patient and residential treatments) for eating disorders (EDs) are associated with high treatment costs and significant challenges for patients and carers, including access difficulties and disruption to daily routines. Intensive community and home-based treatments have been suggested as alternatives to institutionally based intensive treatments for other severe mental illnesses, with promising clinical, social and health economic outcomes. The possible advantages of these treatments have been proposed for EDs, but this emerging area of research has not yet been systematically investigated. This scoping review aims to map the available literature on intensive community and home treatments for EDs, focusing on their conceptualisation, implementation and clinical outcomes.

**Methods** This proposed scoping review will follow the Preferred Reporting Items for Systematic Reviews and Meta-Analyses Protocol extension for Scoping Reviews checklist and the Joanna Briggs Institute Reviewer's Manual. This review will include any peer-reviewed study concerning intensive community and home-based treatments for any EDs, with no restrictions on geographical context or study design. Grey literature will also be considered. The literature search will be conducted in four databases: PubMed, PsycInfo, MEDLINE and Web of Science. Two researchers will independently screen the titles, abstracts and text of the returned articles for eligibility. Data charting and analysis will consist of a narrative description of the included studies, quantitative and qualitative findings relative to the aims of this scoping review. Gaps in the literature will be highlighted to inform future research, clinical practice, and policy.

**Ethics and dissemination** Ethical approval is not required as all data are available from public sources. The results of this scoping review will be disseminated through peer-reviewed publication, conference presentation, and social media.

## STRENGTHS AND LIMITATIONS OF THIS STUDY

⇒ This scoping review will be the first synthesis of published and unpublished literature on intensive community and home treatments for eating disorders.
⇒ This study will be conducted according to standardised guidelines for scoping reviews.
⇒ The use of broad evidence sources and methodologies will increase the likelihood of the identification of all relevant studies.
⇒ Screening and charting will be conducted by two independent researchers to ensure reliability.
⇒ The studies screened and included in this review will be limited to publications in the English language.

## INTRODUCTION

Eating disorders (EDs) are serious and complex conditions associated with significant psychological, social and economic burdens to the person, their carers and society.[1–3] Psychological comorbidities are the norm, prominently including depression and suicidality.[4 5] Likewise, medical complications, secondary to starvation and/or other ED symptoms (eg, purging) are common.[6–8] Unsurprisingly, therefore, mortality rates for EDs, either due to physical health problems or due to suicide, are elevated relative to the general population and other psychiatric illnesses.[1 9] The highest rates of mortality are observed for anorexia nervosa (AN), where risk is more than five times greater when compared with age-matched and gender-matched general populations.[2 10] Accordingly, there is a need for treatment to begin promptly to minimise chronicity and improve the chances of a full recovery.[1 11]

The demand for access to specialist services and intensive ED treatments seems to be increasing.[12] Evidence indicates that while hospital admissions for other mental disorders have either stabilised or decreased worldwide, a different trend is apparent for EDs.[13–15] For instance, in England hospital admissions for EDs doubled between 1998 and 2020 while admissions for other severe mental disorders decreased.[16] This situation appears to have been further exacerbated by the ongoing COVID-19 pandemic. Internationally, research has indicated a rise in new ED presentations, increases in symptom severity and high levels of deterioration, relapse and comorbidities in those with

established EDs.[12 17 18] Unsurprisingly therefore, the need for ED treatment has increased, including the need for ED-related hospitalisations and other forms of intensive treatments for both children and adults.[19–21]

Treatments for EDs are typically offered within a stepped-care model, where the intensity of treatment is matched to the severity of patient presentation. International guidelines recommend that the majority of ED patients should be offered specialist outpatient psychotherapy at the earliest opportunity. The recommended length of such outpatient psychotherapy is around 20 once-weekly sessions for bulimia nervosa (BN) or binge eating disorder (BED) and 20–40 sessions for those with AN.[22] For those who do not adequately respond to outpatient care or whose physical health significantly deteriorates, specialised intensive treatments are recommended.[22–24] Common intensive treatment modalities include inpatient, day patient (also known as partial hospitalisation) and residential treatments. These intensive treatment settings share goals and structures: treatment is multidisciplinary, and patients in either treatment setting receive meal support, nutritional education, and individual and group psychotherapy.[25] The key difference between the settings is that day patient treatment allows for patients to return home in the evening and on weekends.

Clinical guidelines for ED inpatient admission length and discharge criteria vary across EDs and countries. Decisions surrounding the length of hospital or inpatient stay for ED are generally complex, particularly for AN due to the associated psychiatric and medical complications. Data from specialised ED inpatient units in Scotland showed that the mean length of stay (LoS) was about 16 weeks for adults and 20 weeks for adolescents.[26] A retrospective observational study using hospital admission data between 2000 and 2014 in Portugal showed that the median LoS was longest for AN, followed by unspecified ED and BN. Moreover, patients with AN had the highest readmission rates and hospitalisation costs.[27] Recently, Kan and colleagues[28] conducted a comprehensive systematic review and meta-analysis on the LoS for AN. Their findings have demonstrated that the internationally pooled mean LoS in specialist inpatient services was about 11 weeks. However, the pooled mean LoS for AN appeared to be much longer in Europe (~15 weeks) than the USA (~7 weeks) and the other countries (~12 weeks), which might be a result of differences between healthcare systems across countries. Furthermore, age and body mass index (BMI) were found to be significant moderators of the pooled mean LoS in both treatment settings, indicating that the higher the age and lower the BMI, the longer the stay. However, the overall LoS has decreased over time.

Day patient treatment is typically delivered 6–10 hours per day and 3–7 days per week depending on patients' needs and the resources of the services.[29–31] The typical treatment duration mainly falls between 10 and 16 weeks across EDs[32]; consistent with this, the international pooled mean LoS for AN is around 12 weeks.[28] However, LoS can actually vary between 3 and 39 weeks across EDs. For instance, the mean day patient treatment LoS is longer for AN than avoidant/restrictive food intake disorder, though the mean LoS is similar per diagnosis between inpatient and day patient services.[28 33] Thus, it appears that day patient treatment programme structures vary widely between services depending on available resources, guiding theoretical models, treatment aims and symptom severity.[31 34]

Evidence demonstrates that inpatient treatment is associated with the highest costs of any ED treatment,[25] consistently found to be more costly than both day patient[29 35 36] and specialist outpatient treatment.[37] High inpatient treatment costs are especially relevant to those with AN, as their often high medical risk and the time it takes to refeed those with very low BMIs requires a longer LoS.[28 38 39] For instance, research conducted in Germany has demonstrated that over 50% of the yearly treatment costs for AN are due to hospitalisation.[40] Though inpatient treatment costs are most pronounced in AN, the healthcare usage and treatment costs for BN and BED are also higher than those of the general population.[41 42] For example, a recent paper[43] reported a near fourfold increase in mean healthcare spending among patients with EDs ($29456) compared with those without EDs ($7418) in the USA in 2016. While day patient treatment is less costly than inpatient, a recent scoping review suggests that it is less cost-effective than outpatient treatment for those with EDs.[44]

Although common intensive ED treatments, such as inpatient and day treatment, clearly have benefits for patients, including a focus on recovery, regular meal support and improving physical health, they are also likely to bring additional challenges. For instance, inpatient treatment is associated with difficulties in maintaining relationships with friends and family, a sense of disconnection from real life (ie, the potential for patients becoming institutionalised is high), and causes disruptions to education or work routines.[45–47] In addition, the risk of relapse during the first months post-discharge is high.[48–50] In contrast, day patient treatment allows patients to better maintain their relationship to the real world while receiving intensive treatment. However, just like inpatient treatment, attending a day patient programme and thus being at the hospital 3–7 days a week has the potential for disrupting education, employment and leisure activities.[51] In addition, similar unintended iatrogenic effects as in inpatient treatment may be found in day programmes, in that strict schedules and routines around eating and predictable menus may produce an environment that fosters patients' inflexibility and reduces personal agency and self-efficacy.[44] Moreover, outside urban areas with good transport connections, the daily commute to a day patient service can be challenging or impossible.[52 53] For example, the Mental Welfare Commission has reported that in Scotland, day patient services are geographically inequitable, leading to particular difficulties in accessing support in the community for those living in rural areas.[53]

İnce B, *et al. BMJ Open* 2023;**13**:e064243. doi:10.1136/bmjopen-2022-064243

Finally, day patient treatment can also place significant burdens on patients' families, for example, through disruptions to their daily routines and them having to carry increased responsibility for acutely unwell patients' physical and mental health.[54 55] All in all, traditional institutionally based intensive treatments such as inpatient and day treatment are costly, have limited availability and are associated with a number of disadvantages for patients. Additionally, a recent report by BEAT,[56] the UK's leading ED charity for patients and carers, highlighted that patients and families see intensive community and home-based treatment modalities as more acceptable and empowering alternatives to traditional inpatient treatment. Therefore, alternative, and more flexible intensive treatment options that better align with patient preferences need to be explored to improve patient and carer outcomes and treatment experiences, manage increased demand for intensive treatments, and optimally employ scarce resources.

Intensive community (ie, treatment, care and support offered within a patient's local environment in outpatient or community settings) and home-based (ie, treatment, care and support offered within a patient's home) treatments are considered as alternatives to inpatient and day patient treatments. The benefits of these treatments have previously been explored for other severe mental disorders, such as psychosis and mood disorders, in response to the significant challenges associated with traditional institutional care (eg, high treatment costs, long waiting periods, disruptions to daily life, institutionalisation). These treatments were designed to increase direct and easy access to care within an everyday life context (eg, home, community centres and clinics) and minimise hospital (re-)admissions and days spent in inpatient units.[57–59] Intensive community and home-based treatments are usually carried out by a multidisciplinary team and include several clinical contacts per week with flexibility in their provision, according to clinical need. Investigations of intensive community and home-based treatments have provided evidence for their effectiveness and usefulness for managing mental disorders (eg, schizophrenia and other forms of psychosis, personality disorders) that previously would have been treated within an institutional setting. For instance, a naturalistic study on patients with acute mental disorders found that intensive home-based treatment is an effective and cost-effective alternative to inpatient treatment for reducing acute psychotic and depressive symptoms as well as clinical and functional impairment.[60] Similarly, in a mixed psychiatric population intensive home-based treatment improved psychiatric symptoms, suicidality, general functioning and caregiver burden.[61] A randomised controlled trial of adolescents presenting with self-harm and admitted to inpatient services, compared inpatient treatment as usual to early discharge followed by intensive community care. Overall outcomes were similar in both groups, but patients allocated to intensive community care were significantly less likely to report multiple episodes of self-harm.[62] Furthermore, these care models have been shown to be a valuable alternative to inpatient treatment through avoiding hospital admissions,[63] thereby decreasing hospital days and readmission rates.[62 64] Overall, the evidence suggests that these treatment modalities are cost-effective, and have been to linked to positive clinical and social outcomes as well as rates of patient and carer satisfaction comparable to institutionally based intensive treatments.[65–68] Additionally, the community-based nature of these treatment modalities may improve accessibility and circumvent some of the barriers to care outlined previously.

The possible advantages of intensive community and home-based approaches have been proposed for the treatment of EDs,[69–71] however, no attempt has yet been made to systematically review this area of research. We therefore aim to map literature on intensive community and home treatments for EDs through focusing on their conceptualisation, implementation and clinical outcomes. We further aim to identify knowledge gaps and provide directions for future research.

Research questions

1. What is the extent of the available literature on intensive community and home-based treatments for EDs?
2. How are intensive community and home-based treatments for EDs conceptualised and implemented?
3. What is the evidence on the efficacy, acceptability and cost-effectiveness of intensive community and home-based treatments for EDs?

## METHODS AND ANALYSIS

A scoping review will be conducted to map out the literature on intensive community and home-based treatments for ED. This study will follow the Preferred Reporting Items for Systematic Reviews and Meta-Analyses Protocol extension for Scoping Reviews checklist[72] and the Joanna Briggs Institute (JBI) Reviewer's Manual.[73]

### Eligibility criteria

#### Population

This review will consider studies in patients with any type of ED diagnosis with any age and gender, including those with comorbid psychiatric diagnoses.

#### Concept

This review will consider any studies investigating intensive community and home-based treatments for EDs. We define intensive community treatment as any treatment offering more than two therapeutic contacts per week, excluding mere physical monitoring contacts. This intensity cut-off was used as some outpatient therapies for EDs initially offer 2 sessions per week.

#### Context

This review will consider studies conducted in any geographical context or any community or home setting.

#### Types of evidence source

This review will consider studies with all types of research designs (eg, randomised controlled trials, clinically

controlled studies, quasi-experimental and case–control studies, observational studies, case studies) and studies using quantitative, qualitative or mixed-methods. Ongoing clinical trials listed in trial registries will also be considered in order to explore further how intensive community and home treatments are being conceptualised and studied. Evidence from the grey literature will also be considered. Only studies published in English will be included.

Studies will be excluded if they (i) include mixed psychiatric populations, (ii) describe non-intensive treatments (ie, offering fewer than three therapeutic sessions per week) and (iii) describe institution/hospital-based treatments (ie, inpatient and day patient).

### Search strategy
The literature search will be conducted on four main databases: PubMed, PsycInfo, MEDLINE, Web of Science. Sources of unpublished studies and of grey literature will include Scopus, Google Scholar, US ClinicalTrials.gov, WHO International clinical trials registry platform search portal, ISRCTN Registry and the European Union Clinical Trials Register. Finally, we will conduct a reference list search on the final included studies to identify any additional relevant studies.

A limited initial search was conducted in PubMed by Bİ in February 2022, to identify relevant keywords and develop a search strategy. Then, a preliminary search strategy was developed by Bİ and MP and reviewed by US. The search strategy will be tailored to each database by Bİ and MP, and the search will be conducted on studies published until 1 February 2023. The search terms will include variations of "feeding and eating disorders", "community treatment", "intensive outpatient" and "home treatment". A pilot search conducted on PubMed, PsycInfo, MEDLINE, Web of Science on 1 July 2022 are presented in online supplemental material.

### Study selection
After completing the searches, references will be uploaded to Endnote citation management software, and duplicates will be removed. Titles and abstracts will be reviewed and remaining full-text citations will be screened for eligibility by Bİ and MP independently. In the case of any discrepancies, the reviewers will meet to discuss relevant studies. If the two reviewers cannot reach an agreement on the eligibility of a study, this disagreement will be resolved through consultation with US. Rayyan software will be used in this process.

### Data charting
The data charting form developed by the present authors will be used to extract the key information relevant to study aims. The draft data charting form is presented in table 1. This form might be updated during the final review process in order to ensure all relevant data are accurately captured.

**Table 1** Draft data charting from

| Data item | Data |
| --- | --- |
| Author(s) | |
| Publication year | |
| Location and country | |
| Study design | |
| Study population and sample size | |
| Intervention type | |
| Intervention length and dosage | |
| Underlying theoretical model for the intervention | |
| Intervention delivery mode | |
| Professional(s) delivering the intervention | |
| Involvement of carers in the intervention | |
| Admission and discharge criteria (if available) | |
| Main findings | |

The data will be charted by Bİ and MP independently for each of the studies, to assess if they meet inclusion criteria. If any disagreement arises, the reviewers will discuss to reach a consensus. US will be consulted in the case of uncertainty or unsolved disagreement.

### Data analysis and synthesis
The detailed information regarding the selection procedure will be presented in a Preferred Reporting Items for Systematic Reviews and Meta-Analyses flow diagram. A summary of the studies using the data charting form will be provided in a table with an accompanying narrative description relating to the research aims of this review. Where relevant, common and distinctive characteristics of the treatment modalities will be identified. The main findings will be considered in terms of a range of physical, psychological, social, and functional outcomes to provide a comprehensive and holistic overview of the clinical outcomes. The strength of the evidence will be examined using the JBI Levels of Evidence for Effectiveness: level 1 (high)—experimental designs, level 2—quasi-experimental designs, level 3—observational–analytic designs, level 4—observational–descriptive and level 5 (low)—expert opinion. To guide clinical applications, future research and policy makers, knowledge gaps will be highlighted.

### Patient and public involvement
No patients or members of the public were involved in the development of this protocol or will be involved in the final scoping review. Nevertheless, including qualitative studies should give a valuable insight into patient and carer perceptions and experiences with intensive community and home treatments.

## ETHICS AND DISSEMINATION

Ethical approval will not be required for this review as the data will be obtained from publicly available sources. Dissemination of the review's findings will be performed through peer-reviewed publication and through conference presentations. Additionally, we will use social media (ie, Twitter and Instagram) to reach the wider public.

## Summary and conclusion

To our knowledge, this will be the first review to map out the literature on non-institutional (ie, community and home-based) intensive treatments for ED. The findings will provide an overview on the extent, characteristics, clinical outcomes, and cost-effectiveness of these non-institutionally based intensive treatments. Additionally, this review will provide insights into patient and carer experiences with these interventions. We expect that this review will suggest future directions for designing new intensive treatment services or improving the existing ones, as well as for developing research priorities and policies.

**Contributors** US conceived of the idea for this scoping review. Bİ developed the research questions, search strategy, data charting tool and data analysis plan with support from MP. Bİ and MP drafted the manuscript. US supervised the protocol development and critically reviewed the manuscript. All authors have read and approved final version of the manuscript.

**Funding** US receives salary support from the National Institute for Health Research (NIHR) Biomedical Research Centre for Mental Health, South London and Maudsley NHS Foundation Trust and Institute of Psychiatry, Psychology and Neuroscience, King's College London.

**Competing interests** None declared.

**Patient and public involvement** Patients and/or the public were not involved in the design, or conduct, or reporting, or dissemination plans of this research.

**Patient consent for publication** Not applicable.

**Provenance and peer review** Not commissioned; externally peer reviewed.

**ORCID iD**
Başak İnce http://orcid.org/0000-0003-1177-3490

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
