## [Reviewer comments · BMJ Open]

ARTICLE DETAILS

TITLE (PROVISIONAL)	Intensive Community and Home-Based Treatments for Eating Disorders: A Scoping Review Study Protocol
AUTHORS	Ince, Başak; Phillips, Matthew; Schmidt, Ulrike

VERSION 1 – REVIEW

REVIEWER	Pauli, Dagmar University of Zurich, Child and Adolescent Psychiatry
REVIEW RETURNED	07-Jul-2022

GENERAL COMMENTS	This is a very interesting scoping review. Thank you for addressing this new and important area of eating disorder treatment. I have only a few small remarks for the attention of the authors: (Since the page numbers in the header and footer differ, I give both page numbers in each case) 1. page 5/6 line 25-36: LoS between countries might also differ because of economical aspects/different covering of health insurance for hospital stays2. page 6/7 line 8-12: It is not surprising that treatment costs of any medical condition are higher than costs in the general population. How much higher are they in this case?3. page 7/8 line 1: The term "intensive community treatment" is used frequently in this paper. Please define. Is it the same as home-based treatment?4. page 7/8, line 33: Which severe mental health problems? Including ED?5. page 7/8, line 40: Valuable alternative for which diagnoses? Schizophrenia or others?6. S 9/10, line 16: non- intensive treatments: please specify (I assume you are referring to the above definition of < 3 sessions?)7. table 1: I suggest adding two data items: a) underlying theoretical model (eg CBT, CBT-E, FBT, others, none...) b) Involvement of caregivers in intervention (intensive, non-intensive, none) Thank you again for this important work. I look forward to the review.
--

REVIEWER	Anderson, Stephen NHS Greater Glasgow and Clyde, Adult Eating Disorders Service
REVIEW RETURNED	09-Oct-2022

GENERAL COMMENTS	This will be a very valuable contribution to the eating disorder field. We noted in the SIGN Guideline on Eating Disorders in January 2022 a recommendation around research on healthcare models of
---

	eating disorders because of the lack of good evidence to guide development of models of intensive treatment. My only comment relates to the mention of ARFID on Page 5 without stating what these abbreviations mean.
--	---

REVIEWER	Duffy, Fiona The University of Edinburgh
REVIEW RETURNED	18-Oct-2022

GENERAL COMMENTS	Overall this is a timely and very clinically relevant piece of work. The introduction is thorough, I do wonder about the need to also stress the importance of patient choice and also the need for a range of alternatives to choose from, e.g. the introduction of an outreach service should never been seen as an optioning the absence of other models e.g inpatient, but that each compliment each other Methodology - concept -is there potential to include mixed psychiatric population teams where discrete ED outcomes can be separately assessed or will this dilute the purpose of the review. Goes back to the idea of rurality and clinical utility with many services being unable to support specific ED only outreach teams.. Any date limitations to the search process? Is there an option to be more specific in some of the outcomes that might be collated here e.g. I imagine service users will be concerns that only BMI may be considered rather than the full range of psychological, social and functioning outcomes that could be considered. PPI -I wonder if in this section you could include the fact your are actively seeking out qualitative studies on experience as part of your scoping? Minor Typing/grammar: paragraph 5 of introduction:For instance, the mean day patient treatment LoS is longer for AN than ARFID, though the mean LoS similar per diagnosis between inpatient and day patient services No space (might just be way has been transferred over) In addition, similar unintended iatrogenic effects as in inpatient treatment may be found in day programmes, in that strict schedules and routines around eating and predictable menus may produce an environment that fosters patients' inflexibilityand reduces personal agency and self-efficacy (43) Overall, very happy to accept, I am assuming a search of PROSEPRO for competing protocols has already taken place, if not this may be helpful prior to publication.
--

VERSION 1 – AUTHOR RESPONSE

Reviewer: 1

Comments to the Author:

Comment 1: page 5/6 line 25-36: LoS between countries might also differ because of economical aspects/different covering of health insurance for hospital stays

Response: *Thank you for highlighting this issue. We have included a brief statement about the potential impact of health insurance coverage on length of hospital stay on page 5, which reads as follows: “However, the pooled mean LoS for AN appeared to be much longer in Europe (~15 weeks) than the United States (~7 weeks) and the other countries (~12 weeks), which might be a result of differences between healthcare systems across countries.”*

Comment 2: page 6/7 line 8-12: It is not surprising that treatment costs of any medical condition are higher than costs in the general population. How much higher are they in this case?

Response: *We have now provided information on the annual healthcare cost differences between individuals with and without eating disorders on page 6, paragraph 1 where we say: For example, a recent paper (43) reported a near-fourfold increase in mean healthcare spending among patients with eating disorders (\$29,456) compared to those without eating disorders (\$7,418) in the United States in 2016.*

Reference: *Presskreischer, R., Steinglass, J. E., & Anderson, K. E. (2022). Eating disorders in the US Medicare population. *International Journal of Eating Disorders*, 55(3), 362-371.*

Comment 3: page 7/8 line 1: The term "intensive community treatment" is used frequently in this paper. Please define. Is it the same as home-based treatment?

Response: *Thank you for asking us to clarify these concepts. The term ‘intensive community treatment’ refers to the provision of treatment, care and support offered within a patient’s local environment in outpatient or community settings, while home-based treatment provides treatment, care and support offered within a patient’s home. To reflect the differences, we have provided brief definitions for intensive community and home-based treatments (page 7), as follows: Intensive community (i.e., treatment, care and support offered within a patient’s local environment in outpatient or community settings) and home-based (i.e., treatment, care and support offered within a patient’s home) treatments are considered as alternatives to inpatient and day patient treatments. The benefits of these treatments have previously been explored for other severe mental disorders, such as psychosis and mood disorders, in response to the significant challenges associated with traditional institutional care (e.g. high treatment costs, long waiting periods, disruptions to daily life, institutionalisation).*

Comment 4: page 7/8, line 33: Which severe mental health problems? Including ED?

Response: *The study we have cited (Ougrin et al., 2021) did not specify the diagnoses but mentioned that patients were presenting with self-harm. We have now rephrased this misleading statement in our manuscript (page 7) as follows: “A randomised controlled trial of adolescents presenting with self-harm and admitted to inpatient services, compared inpatient treatment as usual to early discharge followed by intensive community care.”*

Reference: *Ougrin, D., Corrigan, R., Stahl, D., Poole, J., Zundel, T., Wait, M., ... & Taylor, E. (2021). Supported discharge service versus inpatient care evaluation (SITE): a randomised controlled trial comparing effectiveness of an intensive community care service versus inpatient treatment as usual for adolescents with severe psychiatric disorders: self-harm, functional impairment, and educational and clinical outcomes. *European Child & Adolescent Psychiatry*, 30(9), 1427-1436. <https://doi.org/10.1007/s00787-020-01617-1>*

Comment: 5. page 7/8, line 40: Valuable alternative for which diagnoses? Schizophrenia or others?
Response: We have provided more details regarding the diagnostic categories that would normally be treated within an institutional setting (page 7), as follows: Investigations of intensive community and home-based treatments have provided evidence for their effectiveness and usefulness for managing mental disorders (e.g., schizophrenia and other forms of psychosis, personality disorders) previously would have been treated within an institutional setting.

Comment: 6. S 9/10, line 16: non- intensive treatments: please specify (I assume you are referring to the above definition of < 3 sessions?)

Response: *We have provided a definition for non- intensive treatments (page 9) as follows “non-intensive treatments (i.e., offering fewer than three therapeutic sessions per week)”.*

Comment: 7. table 1: I suggest adding two data items: a) underlying theoretical model (eg CBT, CBT-E, FBT, others, none...) b) Involvement of caregivers in intervention (intensive, non-intensive, none)

Response: *Thank you for this suggestion. We have added the suggested items into our draft data charting form in Table 1 (page 11) as follows:*

Table 1 Draft Data Charting Form

Data Item	Data
Author(s)	
Publication Year	
Location & Country	
Study Design	
Study Population and Sample Size	
Intervention Type	
Intervention Length & Dosage	
Underlying Theoretical Model for the Intervention	
Intervention Delivery Mode	
Professional(s) Delivering the Intervention	
Involvement of Carers in the Intervention	
Admission & Discharge Criteria (if available)	
Main findings	

Reviewer: 2

My only comment relates to the mention of ARFID on Page 5 without stating what these abbreviations mean.

Response: *Thank you for reviewing our manuscript so positively. We have now spelled out the meaning of the term ARFID before using the abbreviation. (page 5)*

Reviewer: 3

Comment: Overall this is a timely and very clinically relevant piece of work. The introduction is thorough, I do wonder about the need to also stress the importance of patient choice and also the

need for a range of alternatives to choose from, e.g. the introduction of an outreach service should never been seen as an optioning the absence of other models e.g inpatient, but that each compliment each other.

Response: *We have now added a point to the introduction about how these novel options may give patients a range of options to choose from (page 7). This reads as follows: Therefore, alternative, and more flexible intensive treatment options that better align with patient preferences need to be explored to improve patient and carer outcomes and treatment experiences, manage increased demand for intensive treatments, and optimally employ scarce resources."*

Comment: Methodology - concept -is there potential to include mixed psychiatric population teams where discrete ED outcomes can be separately assessed or will this dilute the purpose of the review. Goes back to the idea of rurality and clinical utility with many services being unable to support specific ED only outreach teams.

Response: *Thank you for this suggestion. We initially decided to exclude mixed psychiatric populations in order to map conceptualisation and implementation of intensive community and home treatments that were specifically designed for patients with eating disorders. We understand why including mixed psychiatric populations might be of value, but feel that the specificity of the review to those with eating disorders is a key way in which the paper adds value to the field.*

Comment: Any date limitations to the search process?

Response: *We initially planned to review studies published until 1st June 2022. However, to be mindful of the duration of the peer-review process, we have decided to extend the time frame to studies published until 1st February 2023 (page 9). We have no limitation for the start date.*

Comment: Is there an option to be more specific in some of the outcomes that might be collated here e.g. I imagine service users will be concerns that only BMI may be considered rather than the full range of psychological, social and functioning outcomes that could be considered.

Response: *Thank you for this suggestion. We have not listed any specific outcome measures as we hope to provide a comprehensive and holistic overview of the clinical outcomes used in the studies. We have now highlighted this in the data analysis and synthesis section (page 11) as follows: "The main findings will be considered in terms of a range of physical, psychological, social, and functional outcomes to provide a comprehensive and holistic overview of the clinical outcomes."*

Comment: PPI -I wonder if in this section you could include the fact your are actively seeking out qualitative studies on experience as part of your scoping?

Response: *Thank you for this suggestion. We have now highlighted the benefit of including qualitative studies on patient and carer experiences on page 11, as follows: "Nevertheless, including qualitative studies should give a valuable insight into patient and carer perceptions and experiences with intensive community and home treatments."*

Comment: Minor Typing/grammar:

paragraph 5 of introduction:For instance, the mean day patient treatment LoS is longer for AN than ARFID, though the mean Lois similar per diagnosis between inpatient and day patient services
No space (might just be way has been transferred over)

In addition, similar unintended iatrogenic effects as in inpatient treatment may be found in day programmes, in that strict schedules and routines around eating and predictable menus may produce an environment that fosters patients' inflexibilityand reduces personal agency and self-efficacy (43)

Response: *Thank you for these comments. We have checked the manuscript and it looks like these writing issues appeared during the transfer.*

Comment: Overall, very happy to accept, I am assuming a search of PROSEPRO for competing protocols has already taken place, if not this may be helpful prior to publication.

Response: Thank you. We did originally conduct a search on PROSPERO to identify any protocols that might potentially overlap with our study. We recently conducted a repeat search, and once again did not identify any review protocol for investigating intensive community and home-based treatments for eating disorders.

Reviewer: 1

Competing interests of Reviewer: none

Reviewer: 2

Competing interests of Reviewer: No competing interests.

Reviewer: 3

Competing interests of Reviewer: I declare no competing interests

Editor(s)' Comments to Author:

Comment: Please move the information about the BEAT report to the Introduction where it would be more suitably placed.

Response: As requested, we have moved the information about the BEAT report to the introduction (see page 7, paragraph 1), as follows: "All in all, traditional institutionally based intensive treatments such as inpatient and day treatment are costly, have limited availability and are associated with a number of disadvantages for patients. Additionally, a recent report by BEAT (56), the UK's leading ED charity for patients and carers, highlighted that patients and families see intensive community and home-based treatment modalities as more acceptable and empowering alternatives to traditional inpatient treatment. Therefore, alternative, and more flexible intensive treatment options that better align with patient preferences need to be explored to improve patient and carer outcomes and treatment experiences, manage increased demand for intensive treatments, and optimally employ scarce resources."

Comment: The PRISMA guidelines have been updated and now recommend publishing the full search strategy for each database searched as supplemental material.

Response: We have conducted a further pilot search on 01/07/2022 following our initial submission. We have now included the results of the search strategy for the four main databases as supplemental material.

Additional Request from Editorial Office:

Comment: Kindly remove 'reference number 70' with unpublished data citation. We only include published works in the reference list, although unpublished data/under review can be cited in the main body of the article with the name of the author who wrote the study, and its title.

Response: We have removed the 'reference number 70' with unpublished data citation, as requested by the editorial team following our revision submission.

Reference number 70: Holmes B, Frampton I, Clark-Stone S. Evaluation of the outcomes following home treatment for anorexia nervosa [Unpublished audit report]. 2018.

Comment: Reference citation missing

The in-text citation for 'reference number 6' and 'reference number 52' are missing. Please provide the missing citation and ensure that all citations of references are in ascending order.

Response: We have now corrected the citation mistakes that happened due to an error by the citation manager software. We apologize for this mistake.